# Thickening Properties of Carboxymethyl Cellulose in Aqueous Lubrication

**Jan Ulrich Michaelis** [1,2], **Sandra Kiese** [2,*], **Tobias Amann** [3], **Christopher Folland** [4], **Tobias Asam** [4] **and Peter Eisner** [2,5,6]

1    TUM School of Life Sciences, Technical University of Munich (TUM), D-85354 Freising, Germany
2    Fraunhofer Institute for Process Engineering and Packaging (IVV), Giggenhauser Straße 35, D-85354 Freising, Germany
3    Fraunhofer Institute for Mechanics of Materials (IWM), D-79108 Freiburg, Germany
4    Carl Bechem GmbH, D-58089 Hagen, Germany
5    ZIEL-Institute for Food & Health, TUM School of Life Sciences Weihenstephan, Technical University of Munich, Weihenstephaner Berg 1, D-85354 Freising, Germany
6    Faculty of Technology and Engineering, Steinbeis-Hochschule, George-Bähr-Str. 8, D-01069 Dresden, Germany
*    Correspondence: sandra.kiese@ivv.fraunhofer.de; Tel.: +49-8161-491-525

**Abstract:** Increasingly restricted availability and environmental impact of mineral oils have boosted the interest in sustainable lubrication. In this study, the thickening properties of sodium carboxymethyl celluloses (CMCs) were investigated in order to assess their potential as viscosity modifiers in aqueous gear and bearing fluids. The pressure, temperature and shear dependence of viscosity was studied at different concentrations and molecular weights $M_W$. The tribological properties were investigated at different viscosity grades in both sliding and rolling contact, and compared to rapeseed oil and polyethylene glycol 400. The viscosity of the CMC solutions was adjustable to all application-relevant viscosity grades. Viscosity indices were similar or higher compared to the reference fluids and mineral oil. Temporary and permanent viscosity losses increased with $M_W$. Permanent viscosity loss was highest for high $M_W$ derivatives, up to 70%. The pressure-viscosity coefficients $\alpha$ were low and showed a high dependency on shear and concentration. In rolling contact, low $M_W$ CMC showed up to 35% lower friction values compared to high $M_W$, whereas no improvement of lubricating properties was observed in sliding contact. The results suggest that low $M_W$ CMC has great potential as bio-based thickener in aqueous lubrication.

**Keywords:** cellulose derivative; carboxymethyl cellulose; viscosity modifier; water-containing lubricant; water-based lubricant; aqueous lubrication; biolubrication; biopolymer





## 1. Introduction

The idea of aqueous lubrication is highly tempting, especially given the increasing environmental and political challenges. Water is not only environmentally friendly, locally and globally available, fire resistant and easily disposable, but furthermore, high thermal conductivity and low friction coefficients of water-containing fluids improve the efficiency of tribological systems [1–4]. Unfortunately, the lubricating properties of plain water are rather poor and the insufficient viscosity strongly limits the minimum lubricating film thickness, a decisive factor in risk assessment of sliding wear, particularly in gear transmission [1,4–6].

Low viscosity is generally compensated by the addition of high molecular polymeric additives, commonly referred to as viscosity modifiers (VM) or thickeners. The functionality of water soluble VM depends in particular on thickening performance, viscosity–temperature relationship, shear stability, viscosity-pressure relationship and film-forming capability [6–8]. In comparison to current applications of aqueous lubricants, such

as metal working, requirements in gear transmission are significantly higher. Especially in the contacting area of sliding tooth flanks, thickeners are subjected to high temperatures and extreme shear strain rates [1,9]. Thermosensitive thickeners, such as polyvinyl alcohols [10], polyacrylic acid [11] and polyvinyl pyrrolidone [12], are therefore unsuitable for transmission applications [13]. Polyalkylene glycols provide, in principle, excellent lubricating properties and good biodegradability, but are relatively expensive, susceptible to oxidation and most importantly they are of fossil origin [2,14–16]. Chen et al. [17,18] tribologically investigated lyotropic sugar-based liquid crystals (alkylglucopyranosides) on the macro- and nanoscale. 40% aqueous solution of octyl $\beta$-D-glucopyranoside showed minimum coefficients of friction (COF) of 0.02, which is lower than a standard lubricating oil. One of the disadvantages for large scale applications, however, are the high costs of glucopyranosides.

Polymers of biological origin, often referred to as biopolymers, are mostly non-toxic, biodegradable, eco-friendly and reduce the dependence on non-renewable resources [19]. In combination with water, biological thickeners enable the formulation of sustainable, aqueous lubricants. Especially water-soluble biopolymer derivatives with adjustable, multifunctional properties, such as cellulose or chitosan ether, are a promising alternative to fossil-based polymeric thickeners [20,21]. However, most cellulose derivatives are also thermosensitive, form gels or precipitate when heated. The only exceptions are hydroxyethyl and sodium carboxymethyl cellulose (CMC) [13,22]. Naik et al. [13] studied the potential of hydroxyethyl celluloses as viscosity modifiers in aqueous hydraulic fluids. Benchmarked against standard hydraulic fluids, the resulting wear scar diameters were at least twice as high. In order to improve the lubricating performance, the authors promoted the utilization of lower molecular weight derivatives at higher concentrations. In a more recent study, Gelinski et al. [23] developed and evaluated a hydraulic medium, including glycerol and carboxymethyl chitosan. Although, tribological tests showed improved wear performances at increasing chitosan concentrations, the influence of the polymers on elastohydrodynamic film formation has not been investigated. Sagraloff et al. [4,9] experimentally investigated the influence of different cellulose ethers on the scuffing and wear performance of aqueous gear fluids. Depending on the polymer used, the scuffing load carrying capacity was increased by up to 3 failure load stages. However, due to low elastohydrodynamic film thicknesses of the polymer solutions, all samples showed a high risk of sliding wear.

CMC is a water soluble, anionic, non-toxic, biodegradable, linear polysaccharide of anhydro-glucose, covalently linked by $\beta$-1,4-glycosidic bonds [24]. Carboxymethylation of wood- or cotton-based cellulose with sodium hydroxide-chloroacetic acid results in a partial substitution of protons by carboxymethyl groups ($-CH_2COOH$) [25]. Current application fields of CMC include e.g., the food, textile, cosmetic, pharmaceutical, biomedical and paper industry [24,26]. The rheological behavior of CMC solutions is primarily characterized by their average molecular weight $M_\eta$ and the degree of substitution (DS), the average number of substituted hydroxyl groups per monomer unit [27]. Additionally, uniformity or distribution of substitution is an influencing factor, as less substituted, crystalline segments promote aggregation and thus, thixotropy [28]. For DS above 1.0, crystallinity is close to zero and solubility in water reaches its maximum [29,30]. In the literature, the DS varies widely, going from 0.2 to 2.84 [20,31]; commercially available grades, however, are limited to the range between 0.38 and 1.4 [32]. Polymers with DS below 0.4 are considered as water insoluble [33]. Figure 1 shows the idealized molecular structure of CMC with a DS of 1.0.

In contrast to most cellulose derivatives, CMCs show no thermosensitive behavior and are thermally stable up to 140 °C [34,35]. However, the anionic character of CMC polymers promotes interaction with polyvalent cations and other polyelectrolytes [27]. Guan et al. [36] investigated the anti-wear properties of a fluid containing 2% triethanolamine, 0.5% of a zinc alkoxyphosphate and varying amounts of CMC. Experiments were carried out by means of an MQ-800 four-ball tester at a speed of 1450 rpm and a load of 392 N at 20 °C. The results showed a concentration-dependent improvement of the anti-wear capacity of

the fluid, with an optimum at 0.7 wt.% CMC. To the best knowledge of the authors, the hydrodynamic lubricating properties of aqueous CMC solutions have not been investigated yet. Our aim was to study the rheological and tribological properties of aqueous CMC solutions and investigate the influence of molecular weight, concentration and degree of substitution on elastohydrodynamic film formation. In the first part, shear, temperature and pressure dependence of solution viscosity was investigated. In the second part, the lubricating performance of the CMC solutions was studied at different viscosity grades under sliding and rolling conditions. The results were benchmarked against rapeseed oil and polyethylene glycol 400, two common biodegradable lubricating fluids.

**Figure 1.** Molecular structure of carboxymethyl cellulose (CMC) with a degree of substitution (DS) of 1.0.

## 2. Materials and Methods

### 2.1. Materials

Five types of commercially available CMC were investigated. Table 1 shows the trade name, abbreviation, viscosity range and degree of substitution of all cellulose derivatives. Blanose™ 7ULC and Ambergum™ 1221 were provided by Ashland Industries Europe GmbH (Schaffhausen, Switzerland) and Walocel™ CRT 30, Walocel™ CRT 1000 and Walocel™ CRT 10000 by DuPont de Nemours (Wilmington, NC, USA). Blanose™ 9LCF was provided by Ashland Industries Europe GmbH (Schaffhausen, Switzerland) and used as reference for molecular weight determination.

Disodium hydrogen phosphate, sodium dihydrogen phosphate, ethanolamine, triethanolamine, 2-amino-2-methylpropanol, azelaic acid and glycerol were purchased from Sigma Aldrich (Steinheim, Germany). Polyethylene glycol 400 (PEG) was purchased from Carl Roth (Karlsruhe, Germany). 2-propanol and ethanol were purchased from Th. Geyer (Renningen, Germany). The commercially available rapeseed oil vitaDor (RSO) was supplied by Lidl, Germany. Acticide MBS was provided by Thor (Speyer, Germany). Hordaphos® 145 was provided by Clariant (Pratteln, Switzerland). Gray cast iron chips were purchased from profluid® (Ulm, Germany).

**Table 1.** Trade names, abbreviations, viscosity ranges (at solution concentration) and DS of the CMC derivatives used in this study.

| Trade Name | Abbr. | Viscosity at 25 °C [mPas] | DS |
|---|---|---|---|
| Blanose 7ULC | C7XS | 10–20 (6%) | 0.65–0.90 |
| Ambergum 1221 | C2XS | 10–20 (5%) | 1.15–1.45 |
| Walocel CRT 30 G | C9S | 20–40 (2%) | 0.82–0.95 |
| Blanose 9CLF | B9S | 25–50 (2%) | 0.80–0.95 |
| Walocel CRT 1000 PA | C9M | 550–800 (2%) | 0.82–0.95 |
| Walocel CRT 10000 PA | C9L | 900–1500 (1%) | 0.82–0.95 |

### 2.2. Flow Behavior and Critical Concentration

Sodium phosphate buffer (0.1 M, pH = 7), including 0.1 wt% of Acticide MBS, was mixed with the desired amount of cellulose derivative. To ensure complete homogenization, the solutions were gently stirred using a magnetic stirrer at room temperature for at least 16 h. Prior to all rheological measurements, the CMC solutions were preheated in a water

bath to 40 °C for at least 20 min. Ultrasonification, as documented in other publications, was not applied to avoid possible degradation of the cellulose polymers [37].

The rheological investigations were performed on a Physica MCR 301 rotational rheometer (Anton Paar GmbH, Graz, Austria) using a concentric cylinder measurement system (CC27-SN24807, Anton Paar GmbH) over the shear rate $\dot{\gamma}$ range of 0.1 to 1000 s$^{-1}$. The solutions were prepared and measured at least in duplicate. During all measurements, the temperature $T$ was kept constant at 40.0 ± 0.1 °C and a solvent trap was used to minimize water evaporation. Depending on the rheological behavior, zero-shear viscosities were either calculated by fitting the measurement points to the Herschel-Bulkley or Cross model using OriginPro2018 software. Dilute polymer solutions generally show Newtonian behavior [38]. However, arising turbulences at higher shear rates effected alleged dilatant behavior, found to be depicted more accurately by the Herschel-Bulkley model [39]:

$$\eta = \eta_0 + K\dot{\gamma}^{n-1}, \tag{1}$$

where $\eta$ is dynamic viscosity, $\eta_0$ is zero-shear viscosity, $\dot{\gamma}$ is shear rate, $K$ is flow consistency index and $n$ is flow behavior index. The Cross model (Equation (2)) is used to describe pseudoplastic behavior:

$$\eta_{\dot{\gamma}} = \eta_\infty + [\eta_0 - \eta_\infty]/[1 + (C_{\dot{\gamma}})^P], \tag{2}$$

where $\eta_{\dot{\gamma}}$ is viscosity as a function of shear rate, $\eta_\infty$ is viscosity at infinite-shear rate, $C_{\dot{\gamma}}$ is the cross time-constant or consistency of a solution and $P$ is the (cross-)rate constant [40]. Salt-free solutions of flexible polyelectrolytes can be classified into dilute, semidilute (non-entangled and entangled), and concentrated conditions. The transition concentration $c_e$ from the semidilute entangled to the concentrated region, was determined by plotting zero-shear specific viscosity $\eta_{sp}$ [28],

$$\eta_{sp} = \frac{\eta_0 - \eta_s}{\eta_s}, \tag{3}$$

where $\eta_s$ is the solvent viscosity, as a function of concentration [wt%] on a double logarithmic scale and separately fitting the diluted and concentrated region to the power law equation using OriginPro2018 software [41]:

$$\eta = K\dot{\gamma}^{n-1}. \tag{4}$$

## 2.3. Influence of Pressure

To investigate the effect of pressure on the viscosity and determine the pressure-viscosity coefficient $\alpha$, 10 and 15% of C7XS were mixed with water including 0.1% of Acticide MBS. Measurements were performed on a rotational rheometer (Physica MCR 501, Anton Paar GmbH, Graz, Austria) using a high pressure cell in double gap configuration (DG, Anton Paar GmbH). The inner gap of the measurement system was 0.4 mm and the outer gap 0.44 mm. The amount of liquid used was 8 mL. The viscosity was measured at ambient pressure $\eta_a$ and 200 bar $\eta_p$ at shear rates of 50, 100, 500 and 1000 s$^{-1}$. The temperature $T$ was kept constant at 20 °C during all measurements and the pressure-viscosity coefficient $\alpha$ was calculated using the Barus equation [42]:

$$\alpha = \frac{ln\left(\frac{\eta_p}{\eta_a}\right)}{p - p_a}, \tag{5}$$

where $p$ and $p_a$ are the high (200 bar) and low (ambient) pressure.

## 2.4. Influence of Temperature

To measure the influence of temperature on the solution viscosity at two specific viscosity grades, zero-shear viscosities were set to 46.0 ± 2.3 and 220.0 ± 11.0 mPas by adjusting

the CMC concentrations. Additivation consisted of 1.0 wt% triethanolamine, 0.5 wt% Hordaphos 145 and 0.1 wt% of Acticide MBS. The viscosity as a function of temperature was determined using a concentric cylinder measurement system (CC27-SN24807, Anton Paar GmbH). Measurements were performed at least in duplicate over a temperature range of 10 to 90 °C. The shear rate was kept constant at $50\,\mathrm{s}^{-1}$ during all measurements. The effect of temperature on the dynamic viscosity of CMC solutions was determined by fitting the measurement points to the Arrhenius equation, which is given in its logarithmic form by [43],

$$ln\,\eta = ln\,A - \frac{E_a}{R_G T}, \tag{6}$$

where $A$ is a pre-exponential constant, $E_a$ is the Arrhenius activation energy and $R_G$ is the gas constant. The viscosity-temperature relationship was quantified by adjusting the factor $Q$ to adequate temperatures for aqueous lubrication. The adjusted factor $Q$ was calculated by

$$Q = \frac{\eta_{90\,°C}}{\eta_{30\,°C}}, \tag{7}$$

where $\eta_{90\,°C}$ and $\eta_{30\,°C}$ are the dynamic viscosities at 90 °C and 30 °C.

*2.5. Tribological Investigations*

The tribological performance of the CMC solutions under pure sliding conditions was evaluated by means of a rotational rheometer (Physica MCR 301, Anton Paar GmbH, Graz, Austria) equipped with a ball-on-three-plates tribological cell (Anton Paar GmbH). Figure 2a shows the schematics of the tribological cell. Three plates ($15 \times 6 \times 3\,\mathrm{mm}$) are arranged at a 45° angle to center one ball ($r = 0.00635\,\mathrm{m}$) and evenly distribute the normal load $F_L$. While the plates remain stationary, the ball rotates at a given rotation speed $n$, resulting in a sliding speed $u_s$ in each contact point of $2\pi rn/cos(45°)$ and mean entrainment speed $u_e$ of $u_s/2$ The balls were produced from non-stainless steel 100Cr6 (1.3505) and the plates from stainless steel X5CrNi18-10 (1.4301). Young's modulus $E$ and poisson's ratio $\nu$ were $2.1 \times 10^{11}\,\mathrm{Nm}^{-2}$ and 0.30 for 100Cr6 and $1.8 \times 10^{11}\,\mathrm{Nm}^{-2}$ and 0.24 for X5CrNi18-10, respectively. The initial surface roughness $R_a$ of ball and plates was 30 nm. The solution viscosities were set to $46.0 \pm 2.3$ and $220.0 \pm 11.0$ mPas for each derivative by adjusting the polymer concentrations. In order to prevent excessive wear on the specimen and microbiological growth, all solutions were additivated with 1.0 wt% triethanolamine, 0.5 wt% Hordaphos 145 and 0.1 wt% Acticide MBS. Rheological investigations were performed on a Physica MCR 301 rotational rheometer (Anton Paar GmbH, Graz, Austria) using a concentric cylinder measurement system (CC27-SN24807, Anton Paar GmbH) over the shear rate $\dot{\gamma}$ range of 0.1 to $1000\,\mathrm{s}^{-1}$. During all measurements, the temperature $T$ was kept constant at $40.0 \pm 0.1$ °C. Dynamic viscosity at zero and infinite high shear rate were determined by fitting the measurement values to the Cross' equation (Equation (2)). The maximum temporary viscosity loss (TVL) was calculated by

$$\mathrm{TVL} = (1 - \frac{\eta_\infty}{\eta_0}) \cdot 100. \tag{8}$$

Before each tribological test, the cell and specimen were thoroughly cleaned with isopropanol in an ultrasonic bath. To ensure even temperature distribution, the solutions were preheated to 40 °C for at least 15 min before each measurement. The normal load $F_L$ was set to 10 N, equaling a maximum Hertz contact pressure of 0.63 GPa. Each tribological system was first run-in for 10 min at a constant sliding speed $u_S$ of $500\,\mathrm{mms}^{-1}$. Afterwards, the sliding speed $u_S$ was increased and decreased from 0.1 to 1400 mm/s for 8 consecutive runs. The temperature $T$ was kept constant at $40.0 \pm 0.1$ °C. After completion of the measurement, the wear scar diameters (WSD) on the plates were measured by means of an MZ16 stereo microscope from Leica microsystems (Wetzlar, Germany). Each diameter was

measured at least four times. Friction regimes were classified according to the dimensionless lambda ratio $\lambda$, representing the ratio of minimum lubricating film thickness $h_{min}$ to the composite roughness of the contacting surfaces $R_{a,1}$ and $R_{a,2}$ [44],

$$\lambda = \frac{h_{min}}{\sqrt{R_{a,1}^2 + R_{a,2}^2}}. \tag{9}$$

According to Hamrock and Dowson [45], the minimum film thickness $h_{min}$ in point contact at initial Hertzian pressure is calculated as follows:

$$h_{min} = 3.63 \cdot U^{0.68} \cdot G^{0.49} \cdot R \cdot W^{-0.073}(1 - e^{-0.68k}), \tag{10}$$

$$U = \frac{\eta u_E}{2E'R}, \tag{11}$$

$$G = \alpha E', \tag{12}$$

$$W = \frac{F_N}{E'R^2}, \tag{13}$$

$$F_N = \frac{F_L\sqrt{2}}{3}, \tag{14}$$

where $U$ is the dimensionless speed parameter, $G$ the geometry parameter, $W$ the load parameter, $E'$ the effective modulus of elasticity ($2.11 \times 10^{11}\,\mathrm{Nm^{-2}}$), $R$ is the effective Radius and $k$ the ellipticity of the ball ($\sim$1).

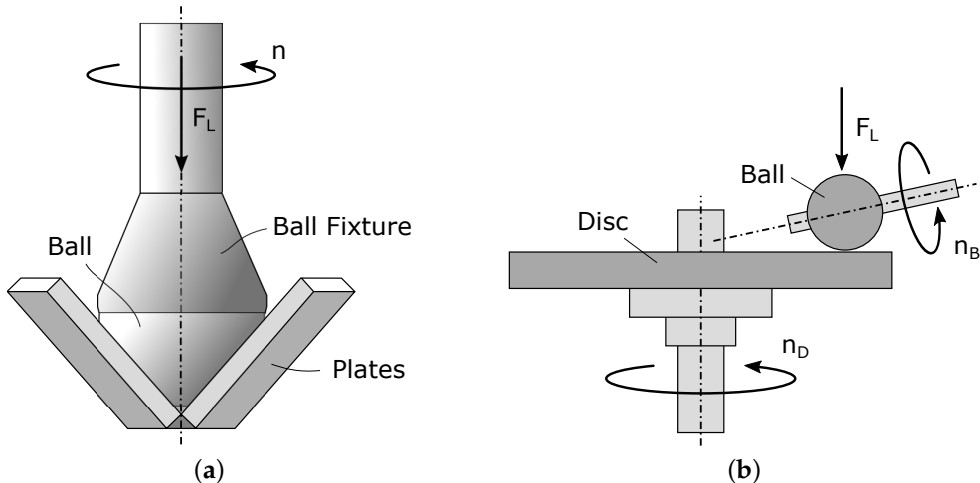

**Figure 2.** Schematic of (**a**) the ball-on-three-plates and (**b**) ball-on-disc tribological setup.

Further testing in rolling contact was performed on a disc-on-ball tribological system (Mini Traction Machine, PCS Instruments, London, UK). Figure 2b shows the structure and operating principle of the measurement system. The slide to roll ratio (SRR) is defined as the ratio of absolute sliding speed $|u_B - u_D|$ to the entrainment speed $u_E = (u_B - u_D)/2$, where $u_B$ and $u_D$ are the surface speeds of the ball and disk, respectively. The coefficient of friction $\mu$ was measured in the velocity range between 1 to 3500 mm/s at a constant temperature of $40.0 \pm 0.1\,^{\circ}$C and normal load of 30 N, equaling a maximum Hertz contact pressure of 0.95 GPa. The SRR was set to 30% during all measurements. The specimen, ball and disc were produced from steel grade 100Cr6, both with a surface roughness $R_a$ of 20 nm. The diameter of the ball was 9.5 mm.

### 2.6. Shear Stability—Permanent Viscosity Loss

The shear stability test evaluated the permanent viscosity loss (PVL) under high shear stress. The test setup and implementation are based on the tapered roller bearing test, a methodology commonly applied in gear oil applications [46]. As shown in Figure 3, the cell consists of a tapered roller bearing (FAG 32008-XDY), case (bearing seat), base plate, top plate and shaft. The cell was sealed by two O-rings and a rotary shaft seal to avoid evaporation and spillage. The case, shaft and both plates were made from steel and manufactured by means of a CNC milling machine in the internal manufacturing. The stress and load, specified in DIN 51350-6 (1450 rpm; 60 °C and 20 h), were adjusted to aqueous lubrication. The lubricant volume was 22 mL. The rotational speed was set to 1000 rpm by a stepper motor engine. The cell temperature was kept at constant $40 \pm 0.1$ °C using a water bath and the axial load was set to 50 N. Solution viscosities were set to $46 \pm 4.6$ mPas for each derivative and all samples were additivated with 1.0 wt% ethanolamine, 0.5 wt% triethanolamine, 0.5 wt% Hordaphos 145 and 0.1 wt% of Acticide MBS. Zero viscosity of the fresh and sheared solutions was determined by means of a parallel plate measurement system (PP50, Anton Paar GmbH) at least in duplicate over a shear rate range of 0.1 to $10{,}000\,\mathrm{s}^{-1}$. The PVL is defined as the decrease in zero-shear viscosity due to shear and was calculated by

$$\mathrm{PVL} = \frac{\eta_{0,f} - \eta_{0,s}}{\eta_{0,s}}, \tag{15}$$

where $\eta_{0,f}$ is the zero viscosity of the fresh (before) and $\eta_{0,s}$ of the sheared (after) solution.

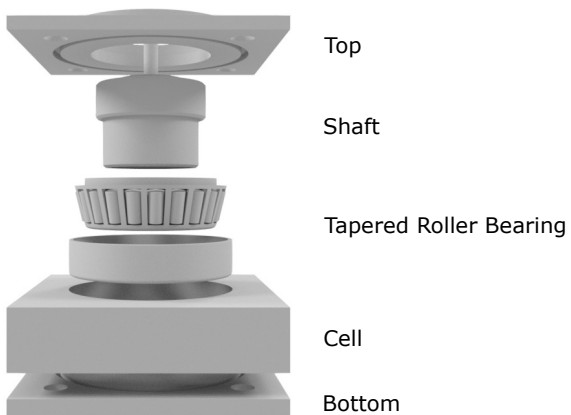

**Figure 3.** Design of the tapered roller bearing shearing cell.

## 3. Results and Discussion

### 3.1. Viscosity-Concentration Relationship

The flow curves were measured in a series of at least eight concentrations ranging from 0.05 to 17.5 wt%, depending on the molecular weight of the CMC. The respective weight percentages were selected to cover the viscosity range between 1 and 220 mPas, corresponding to the relevant viscosity grades of available lubricants for many target applications. Figure 4 exemplarily shows the resulting flow curves for C9L on the left and C2XS on the right. The flow curves of the other derivatives are available in Supplementary Materials. At low concentrations, the flow characteristics were nearly Newtonian (colored in dark gray) for all derivatives. More concentrated solutions (colored in light gray) showed shear-thinning behavior, in which case the curves can be divided into a "Newtonian viscosity plateau" at low, and a shear-thinning part at high shear rates. The beginning of shear thinning behavior, also referred to as critical shear rate $\dot{\gamma}_c$, shifted to lower shear rates with increasing concentration and molecular weight $M_W$, which is in agreement with literature [47].

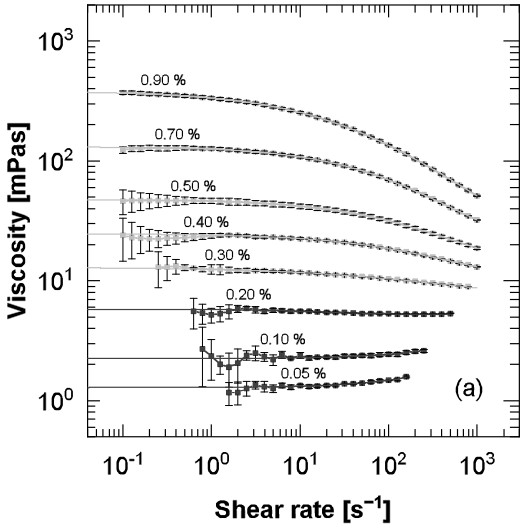
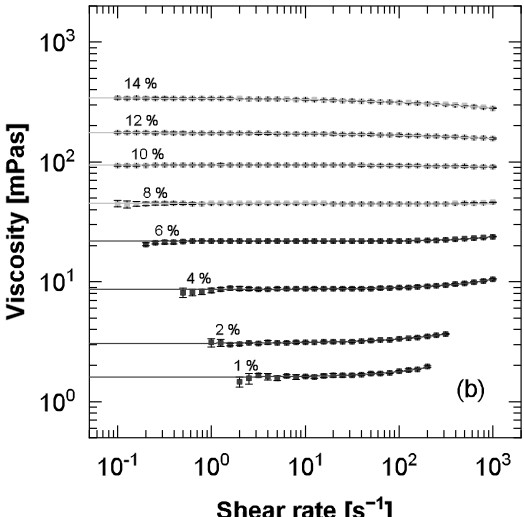

**Figure 4.** Viscosity as a function of shear rate at different concentrations, exemplary shown for (**a**) the high-molecular weight C9L and (**b**) the low-molecular weight derivative C2XS. The plotted points display the measurement results and the lines the fitted curves according to Cross' (colored in light gray) or Herschel's model (colored in dark grey).

The zero-shear viscosities $\eta_0$ were determined by fitting the data of dilute solutions to the Herschel-Bulkley equation (Equation (1)) and of the concentrated, shear-thinning solutions to Cross' equation (Equation (2)). At lower concentrations, inertial effects or turbulences within the measurement gap resulted in a viscosity increase with shear rate and thus, erroneous Newtonian fittings. The effect was successfully compensated by using the Herschel-Bulkley model. All derivatives showed the expected increase in zero-shear viscosity with increasing CMC concentration [47]. At maximum concentration, C7XS was slightly thixotropic in behavior, which is usually effected by low DS [28]. All other solutions showed no time-dependent viscosity changes.

### 3.2. Critical Concentration

Figure 5a shows the specific viscosity at zero-shear rate $\eta_{sp}$ as a function of solution concentration on double logarithmic scale. The lines represent the best fit power laws in the semidilute unentangled and entangled regimes. At $c_e$, polymer chains begin to entangle, which corresponds to the transition from Newtonian to shear thinning behavior [48,49]. Table 2 lists the critical concentrations $c_e$ and the respective zero-shear viscosities $\eta_e$ for all derivatives. Despite the lower average molecular weight $M_W$, the critical concentration of C7XS was lower compared to C2XS, which is attributed to partially unsubstituted microcrystalline parts. Higher DS increases the amount of soluble molecules and thus the solution viscosity [50].

**Table 2.** Average molecular weight $M_W$, degree of substitution (DS), critical concentration $c_e$, critical viscosity $\eta_e$, slope values in the entangled $m_e$ and concentrated regime $m_c$ of the investigated derivatives.

| Derivative | $M_W$ [kg/mol] | DS | $c_e$ [wt%] | $\eta_e$ [mPas] | $m_e$ [1] | $m_c$ [1] |
|---|---|---|---|---|---|---|
| C7XS | 24 * | 0.7 | 8.25 | 13.95 | 1.66 | 4.59 |
| C2XS | 35 * | 1.2 | 6.60 | 21.67 | 1.74 | 3.64 |
| C9S | 88 * | 0.9 | 2.10 | 14.60 | 1.71 | 3.44 |
| B9S | 100 | 0.9 | 1.41 | 7.71 | 1.45 | 3.64 |
| C9M | 240 * | 0.9 | 0.73 | 11.27 | 1.62 | 3.30 |
| C9L | 520 * | 0.9 | 0.27 | 8.17 | 1.54 | 3.10 |

* estimated according to Miehle et al. [48].

In comparison to the literature data, a critical concentration of C9M fairly corresponds to the results of Wagoner et al. [51], who reported a critical concentration of 0.67 wt% for a CMC of 250 kDa (DS 0.7). The resulting zero-shear viscosity of 22 mPas, however, is significantly higher and more comparable to the results for C2XS or C7XS. The same applies to the value of $1.5 \, \text{g L}^{-1}$ reported by Charpentier et al. [52] for a CMC of 300 kDa (DS 0.9). In contrast, values published by Miehle et al. [48], 1.3 wt% and 6.4 mPas for a CMC of 100 kDa (DS 0.9), are clearly below the results for C9S. Overall, $c_e$ is strongly dependent on molecular weight, uniformity of substitution and degree of substitution [48]. As intrinsic viscosity is strongly influenced by ionic strength and thus molarity of the buffer, the same might apply to the level of critical zero-shear viscosity [53,54]. Moreover, the molecular weights provided by manufacturers are sometimes inaccurate, complicating the direct comparison of the results [55].

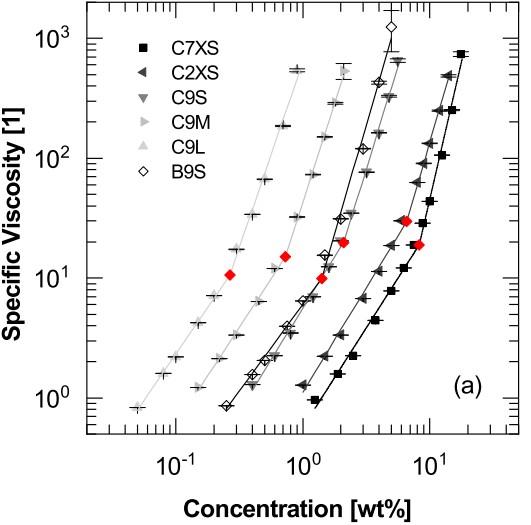 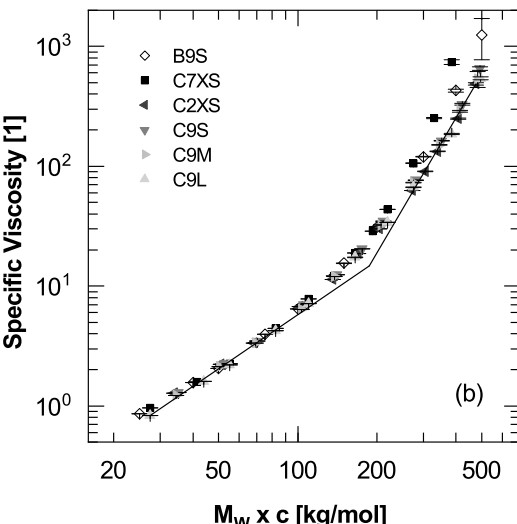

**Figure 5.** (**a**) Double logarithmic plot of the specific viscosity as a function of concentration. Plotted lines correspond to the best fit power laws and red dots to the critical concentrations $c_e$ (**b**) Double logarithmic plot of the specific viscosity as a function of product of viscosity average molecular weight and concentration. Plotted lines show the scaling predictions for CMC solutions of 1.5 and 3.75, published by Lopez et al. [28].

According to Miehle et al. [48], multiplying solution concentration with average molecular weight results in near superimposition of the viscosity graphs. Using B9S, with an average molecular weight of 100 kDa, as given by the supplier, allowed for an estimation of the remaining average molecular weights. Figure 5b shows the resulting superimposition. The estimated average molecular weights are listed in Table 2.

Noticeable in both figures, specific zero viscosity of B9S and C7XS increased faster with concentration compared to the other investigated derivatives. Similar behavior was observed by Barba et al. [56] in relation with interchain interactions of less substituted molecules. All curves showed no sharp transition or kink at the calculated critical concentration, which is similar to literature results [28,48,49,57]. In all publications, smooth transitions are attributed to polydispersity of the polymers. The slope values of all power law fittings for the entangled semidilute $m_e$ and concentrated $m_c$ regions are given in Table 2. The lines in Figure 5b show the scaling predictions for CMC of 1.5 and 3.75, published by Lopez et al. [28]. At high and low solution viscosities, the measured values are in close agreement with the predictions. The slope values of C9L in the semidilute and C2XS in the concentrated region showed the highest correlation. Overall, deviations from the predicted slope values increased with the number of fitted measurement points, as smooth transition and curvature resulted in higher gradients in the entangled and lower gradients in the concentrated region.

With the exception of C7XS and B9S, the critical concentration $c_e$ and the respective zero-shear viscosity clearly decreased with molecular weight. Plotted on a double logarithmic scale, the measurement points of both parameters were in close agreement with linear fits. Equation (16) describes the resulting correlation between $c_e$ and $M_\eta$ and Equation (17) between $\eta_e$ and $M_\eta$.

$$log(c_e) = 2.62087 - 1.17071 \cdot log(M_\eta) \tag{16}$$

$$log(\eta_e) = 1.86543 - 0.34915 \cdot log(M_\eta) \tag{17}$$

The theoretical critical concentration $c_e$ of 50 wt% therefore equals a molecular weight $M_W$ of 6.13 kDa and a critical zero-shear viscosity $\eta_e$ of 38.95 mPas. In conclusion, all solutions with zero-shear viscosities $\eta_0$ above 21.67 mPas and thus fluids of viscosity classes above ISO VG 22 are expected to show shear-thinning behavior. Nevertheless, low molecular weight derivatives showed less shear thinning behavior and are therefore preferable for lubricating applications.

### 3.3. Viscosity-Pressure-Relationship

The pressure-viscosity coefficients of aqueous 10 and 15% C7XS solutions were determined as a function of the shear rate at 20 °C (Figure 6). In each case, dynamic viscosities were measured at 1 and 200 bar and the pressure-viscosity coefficient was calculated using Equation (5).

At the lower concentration of 10%, the pressure-viscosity coefficients $\alpha$ were clearly smaller than at 15% C7XS. However, at 15% $\alpha$ showed a greater dependence on shear rate $\dot{\gamma}$, which corresponds to the flow behavior of the solutions, determined in Section 3.1. Whereas the 10% solution showed near Newtonian flow behavior at ambient pressure and shear rates below $1000\,\text{s}^{-1}$, the 15% solution behaved in a pseudoplastic way. According to Cook et al. [58], the effect of pressure on solution viscosity is comparable to an increase in polymer concentration, indicating that the critical shear rate $\dot{\gamma}_c$ decreases with increasing pressure and the shear-thinning properties become even more prominent. In contrast to most organic solvents, water does not show an exponential increase in viscosity with pressure [58]. The pressure-dependence observed for the CMC solutions is therefore assumed to be entirely effected by the polymers. Addition of CMC increased the pressure-viscosity coefficient $\alpha$ of pure water ($\sim 0.8\,\text{GPa}^{-1}$ [59]) to a maximum of about $3.5\,\text{GPa}^{-1}$ at 20 °C. However, in comparison to rapeseed oil with a coefficient of $14.1\,\text{GPa}^{-1}$ at 25 °C, polyethylene 400 with $\sim 11.8\,\text{GPa}^{-1}$ at 20 °C (calculated from the viscosity data in [60]) and glycerol with $5.9\,\text{GPa}^{-1}$ at 30 °C [61] coefficients of the CMC solutions are clearly lower. Pressure-viscosity coefficients of water-containing polyalkylene glycol-based gear lubricants with water contents of up to 70% are between 5.61 and $6.26\,\text{GPa}^{-1}$ at 40 °C [2]. Wang et al. [62] calculated a coefficient of $6.08\,\text{GPa}^{-1}$ for a 40% polyalkylene glycol solution and Wan et al. [63,64] measured pressure-viscosity coefficients around $\sim 2\,\text{GPa}^{-1}$ for 15% polyalkylene glycol solutions and differing concentrations of propylene glycol at 22 °C, which are slightly higher than the viscosity-pressure coefficients of the C7XS solutions at high shear. In general, highly stressed machine elements, such as gears and rolling bearings, are governed by elastohydrodynamic lubrication and the minimum lubricating film thickness is an important factor in risk assessment [4,65]. As the minimum film thickness in line contact is proportional to $\alpha^{0.54}$ [66], shear-dependence results in a decrease of film thickness with increasing shear stress, counteracting the sliding speed related increase in film thickness. Since the pressure-viscosity coefficient $\alpha$ of aqueous glycerol and polyalkylene glycol solutions decreases with increasing temperature and water content, similar is to be expected for CMC solutions [59,62]. Accordingly, in order to increase the pressure viscosity coefficient $\alpha$, an increase in CMC concentration is required.

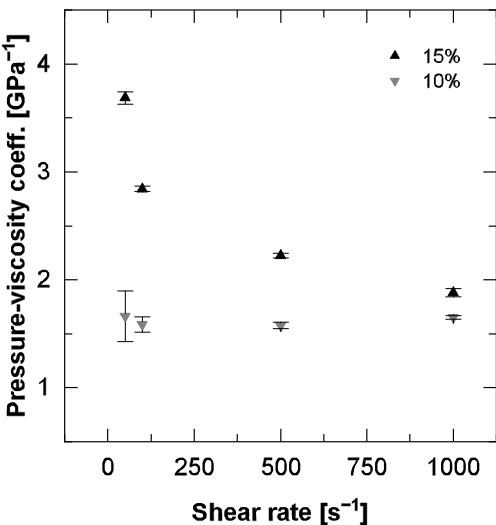

**Figure 6.** Viscosity -pressure coefficient as a function of shear rate for aqueous 10 and 15% C7XS solutions at 20 °C.

### 3.4. Viscosity-Temperature Relationship

Figure 7a shows the viscosity as a function of temperature in a semi-logarithmic plot for C7XS, C9S and C9L, at zero-shear viscosity grade (ZSVG) 46 and 220 mPas and a shear rate of $50^{-1}$. Due to the similarity of the measurement results with C9S, the curves for C2XS and C9M are not shown. Figure 7b shows dynamic viscosity as a function of temperature for PEG and RSO. Table 3 lists the temperature related viscosity losses in terms of $Q$ for all tested solutions.

At temperatures below 20 °C, all samples showed running-in behavior, caused by uneven heat distribution inside the measuring system. Overall, the dynamic viscosities of all fluids, CMC solutions and references, decreased with increasing temperature and the Arrhenius equation (Equation (6)) could be adapted to the measurement points above 30 °C. Lower viscosities of C9L at the starting temperature of 10 °C can be explained by shear-thinning flow behavior. As can be seen in Figure 4a, the viscosities of aqueous C9L solutions at shear rates of $50\,\mathrm{s}^{-1}$ are clearly lower than the zero-shear viscosities. The solutions of the highest molecular weight derivative C9L were the least affected by changes in temperature, with a maximum $Q$ factor of 0.252. The $Q$ factors of C2XS, C9S and C9M at zero solution viscosity of 46 mPas were between 0.160 and 0.178, which corresponds well to the $Q$ factor of RSO. The viscosity of vegetable oils generally exhibits low temperature dependency and the viscosity indices are more than twice those of mineral oils [67–69]. Thus, the comparability of $Q$ indicates excellent temperature behavior. Especially in applications with fluctuating temperatures, fluids with higher $Q$ values are preferable in order to ensure proper operation [7,46]. By using the Arrhenius fit functions and estimating the densities of the aqueous solutions to $0.99\,\mathrm{g\,cm}^{-3}$ at 40 °C and $0.96\,\mathrm{g\,cm}^{-3}$ at 100 °C [70], the kinematic viscosities at 40 and 100 °C were calculated, leading to viscosity indices between 160 and 305. In comparison, viscosity indices of concentrated hydroxyethyl cellulose solutions, measured by Naik et al. [13], were clearly smaller between 114 and 126. Yilmaz et al. [2] determined viscosity indices between 135 and 189 for water-containing gear fluids, based on polyalkylene glycols.

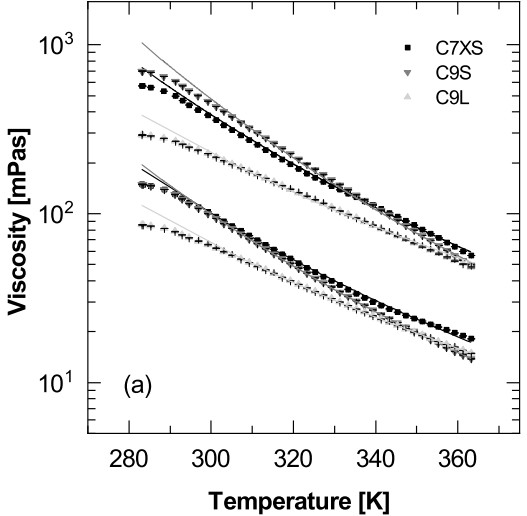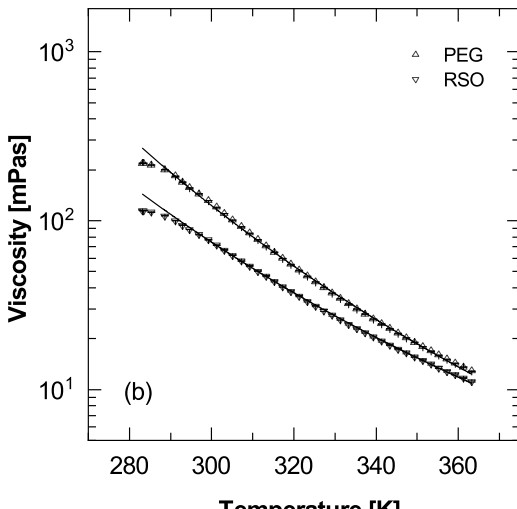

**Figure 7.** Semi-logarithmic plot of the dynamic viscosity as a function of temperature at shear rate $50^{-1}$ for (**a**) the cellulose derivatives C7XS, C9S and C9L at zero-shear viscosity grade 46 and 220 and (**b**) reference fluids PEG and RSO. The plotted points display the measurement results and the lines the fitted curves according to the Arrhenius equation.

**Table 3.** Viscosity-temperature relationship, quantified by the $Q$ factor, for the all tested CMC solutions and reference fluids.

|  | **C7XS** | **C2XS** | **C9S** | **C9M** | **C9L** | **PEG** | **RSO** |
|---|---|---|---|---|---|---|---|
| $Q$ at ZSVG 46 | 0.203 | 0.170 | 0.160 | 0.178 | 0.252 | 0.123 | 0.168 |
| $Q$ at ZSVG 220 | 0.170 | 0.141 | 0.124 | 0.139 | 0.234 |  |  |

Courses of C7XS showed a slight deviation from Arrhenius behavior at higher temperatures, above 70 °C at ZSVG 46 and 50 °C at ZSVG 220, generally indicating changes in structure due to gelation or even precipitation [71]. Small deviations of the C7XS solutions are most likely caused by lower and less homogeneous substitution compared to the other derivatives [50]. Accordingly, in order to avoid temperature-dependent changes in structure and solution viscosity, DS above 0.95 are preferable.

### 3.5. Temporary Viscosity Loss

In order to determine the TVL and analyze the influence of flow behavior, concentration and viscosity on the tribological properties of CMC solutions, dynamic zero-shear solution viscosities were set to the ZSVG 46 and 220.

Figure 8a,b shows the dynamic viscosity $\eta$ as a function of shear rate $\dot{\gamma}$ at 40 °C for ZSVG 46 and 220, respectively, both displayed on a double logarithmic scale. The zero-shear $\eta_0$ and infinite-shear $\eta_\infty$ viscosities were determined by fitting the measurement values to the Cross' equation (Equation (2)). Table 4 shows concentration $c_{CMC}$, zero-shear $\eta_0$ and infinite-shear $\eta_\infty$ viscosity, maximum temporary shear loss (TVL) and wear scar diameter (WSD) for all tested fluids. Adjustment of zero-shear solution viscosity, allows for a direct comparison of flow behavior, infinite viscosity and shearing loss.

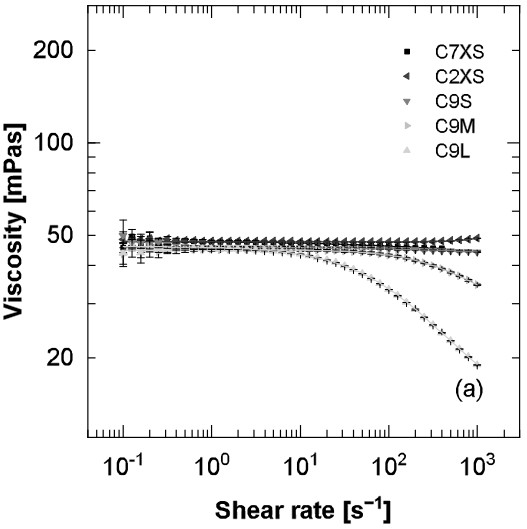
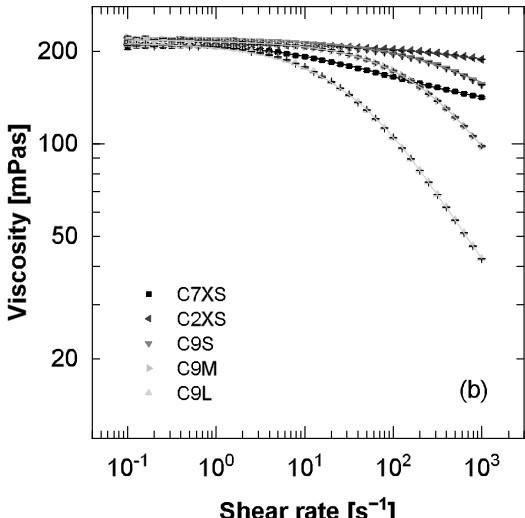

**Figure 8.** Viscosity as a function of shear rate at (**a**) ZSVG 46 and (**b**) ZSVG 220. The plotted points display the measurement results and lines the fitted curves according to Cross' model ($T = 40\,°C$).

Noticeably, the flow behavior of C7XS is clearly different compared to the higher substituted derivatives. Especially in direct comparison to the similar sized C2XS, shear-thinning of C7XS starts at lower shear rates and TVL is significantly higher. Similar behavior has been observed with other, lower substituted derivatives, in preliminary tests. The most likely reason is the aforementioned lower and less homogeneous substitution. Investigations by Kulicke et al. [30,50] showed that CMC molecules with DS below ∼0.95 are only partially solvated, which results in a lower zero viscosity at an equal concentration and molecular weight compared to higher substituted derivatives. Moreover, solutions of C7XS were rather turbid suspensions in appearance than clear solutions, typical for CMC with DS below ∼0.85 [30]. In conclusion, DS around 0.95 are preferable to exploit the highest thickening capacity at equal concentration and average molecular weight.

**Table 4.** Concentration $c_{CMC}$, viscosity at zero $\eta_0$ and infinite $\eta_\infty$ shear rate, temporary viscosity loss TVL, wear scar diameter WSD, minimum lubricating film thickness $h_{min}$ and lambda ratio $\lambda$ for all CMC solutions at ZSVG 46 and 220, the reference fluids and additivated water.

| Derivative | $c_{CMC}$ [wt%] | $\eta_0$ [mPas] | $\eta_\infty$ [mPas] | TVL [%] | WSD [µm] | $h_{min}$ [nm] | $\lambda$ [1] |
|---|---|---|---|---|---|---|---|
| C7XS | 11.670 | 47.90 ± 0.06 | 41.49 ± 1.20 | 13.38 | 482 | 32.10 | 0.76 |
|  | 15.740 | 213.60 ± 0.75 | 129.80 ± 1.69 | 39.23 | 1593 | 74.32 | 1.75 |
| C2XS | 8.210 | 47.63 ± 0.02 | - | 0 | 497 | 35.26 | 0.83 |
|  | 12.940 | 210.06 ± 0.04 | 181.30 ± 6.42 | 13.69 | 1592 | 93.28 | 2.20 |
| C9S | 3.060 | 45.70 ± 0.02 | 43.08 ± 0.03 | 5.73 | 328 | 32.93 | 0.78 |
|  | 4.990 | 219.80 ± 0.20 | 97.89 ± 0.83 | 55.46 | 357 | 61.35 | 1.45 |
| C9M | 1.090 | 44.89 ± 0.02 | 27.70 ± 0.56 | 38.29 | 372 | 24.39 | 0.57 |
|  | 1.870 | 217.19 ± 0.20 | 25.98 ± 1.92 | 88.04 | 547 | 24.89 | 0.59 |
| C9L | 0.415 | 46.07 ± 0.07 | 12.30 ± 0.19 | 73.30 | 301 | 14.04 | 0.33 |
|  | 0.750 | 213.73 ± 0.43 | 14.18 ± 0.69 | 93.37 | 332 | 16.49 | 0.39 |
| AW |  | 0.77 ± 0.00 |  |  | 347 |  |  |
| RSO |  | 32.64 ± 0.05 |  |  | 352 |  |  |
| PEG |  | 45.42 ± 0.01 |  |  | 233 |  |  |

Excluding C7XS from the evaluation, shear losses increased with average molecular weight and ZSVG. Unexpectedly, the flow behavior of C2XS at 46 mPas was near-

Newtonian, if not slightly dilatant, which is in contradiction to the maximum Newtonian zero-shear viscosity of 21.67 mPas, determined in Section 3.2. The upward deviation is attributed to lower salt contents of the solutions, resulting in an expansion of the polyelectrolyte coils and hence a viscosity increase [54]. For the upper ZSVG 220, all solutions showed shear thinning flow behavior. As expected, the highest molecular weight derivative CL9 showed the greatest losses under shear, with a maximum TVL of 93.37%. Differences in shear loss, between C2XS and C9L, were around 80%, regardless of the viscosity grade. Comparable polyalkylene glycol (970 Da) based lubricating oils with water contents of up to 80% showed Newtonian flow behavior at shear rates below $1000\,\text{s}^{-1}$ [62]. As expected, the reference lubricating fluids, PEG and RSO showed Newtonian flow behavior. The measured dynamic viscosities are in good correlation with the literature [60,72] and listed in Table 4. As maximum shear rates in highly loaded machine elements, such as gears or bearings, reach values of up to $10^7\,\text{s}^{-1}$ [73], calculated infinite-shear viscosities $\eta_\infty$ correspond approximately to the dynamic viscosity under application conditions. Infinite-shear viscosities could be increased by increasing the CMC concentration, however, high zero-shear viscosities would considerably reduce the energy efficiency of the lubricated contact and moreover complicate production and handling of the fluids [74]. In order to achieve Newtonian flow behavior at ZSVG 220, a further reduction in molecular weight and an increase in CMC concentration would be required.

*3.6. Lubricating Performance under Sliding Conditions*

The effect of flow behavior on the tribological properties was first studied in pure sliding motion in steel/steel tribological contacts at 10 N normal load and 40 °C. Figure 9a,b show the coefficient of friction (COF) as a function of sliding speed $v$ at ZSVG 46 and 220 respectively, both plotted on a semi-logarithmic scale. The values shown, are the averages and standard deviations of the last three test runs at decreasing velocity. The measured curves are compared to the reference curve of additivated water (AW), including 1.0 wt% triethanolamine, 0.5 wt% Hordaphos 145 and 0.1 wt% Acticide MBS without the addition of CMC, with a Newtonian dynamic viscosity of 0.77 mPas at 40 °C (Figure 9a), and the reference lubricating fluids PEG and RSO (Figure 9b). The resulting wear scar diameter (WSD) of all tested solutions is listed in Table 4. Figure 10 exemplarily shows the optical images of the WSD at ZSVG 46 for C9L, C2XS and C7XS.

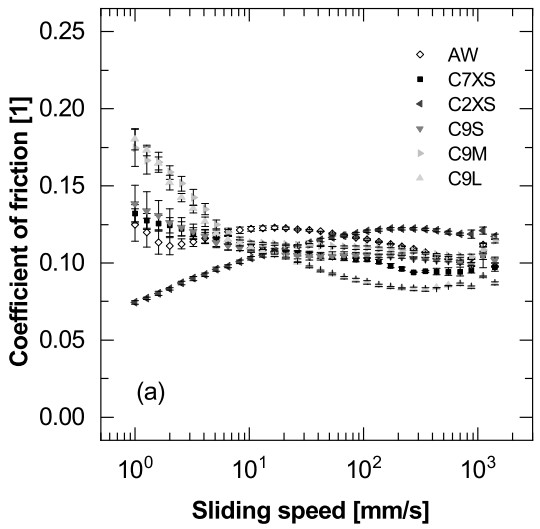
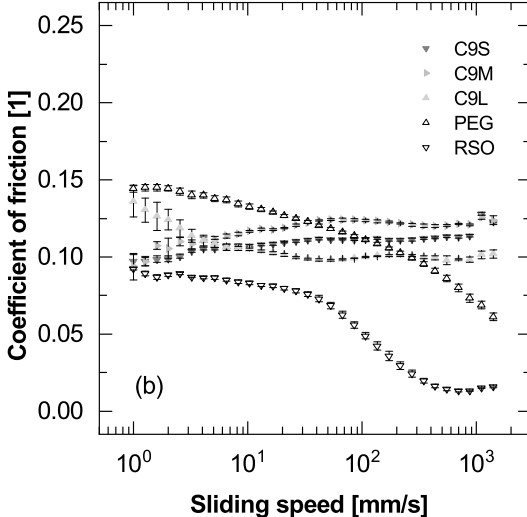

**Figure 9.** COF as a function of sliding speed at (**a**) ZSVG 46 and (**b**) ZSVG 220 (ball-on-three-plates, $F = 10\,\text{N}$, $T = 40\,°\text{C}$).

For AW, the COF was between 0.12 and 0.10. Due to the low viscosity, the contact operates solely in the boundary lubrication regime. At ZSVG 46, the friction curves of the

aqueous CMC solutions varied with concentration and molecular weight. Whereas the COF for C9L decreased with the logarithm of sliding speed from 0.17 to 0.08, the friction values for C2XS increased steadily from 0.08 to 0.12. The curves for C7XS, C9S and C9M showed a decrease in friction at low sliding speeds, but COF stayed almost constant around 0.1 above $10\,\text{mm}\,\text{s}^{-1}$. At ZSVG 220, the measurements for C2XS and C7XS resulted in scuffing. The corresponding curves are not displayed, due high fluctuations of COF. COF for C9M increased initially and reached a constant value of 0.13. The same applies to C9S, though constant friction values were slightly lower, around 0.11. C9L showed a slight decrease at low sliding speeds and stayed nearly constant, around 0.10 at velocities above $10\,\text{mm}\,\text{s}^{-1}$. The COF of RSO was highest at low sliding speeds and showed a sharp decrease in friction at speeds above $50\,\text{mm}\,\text{s}^{-1}$, indicating a hydrodynamic lift. RSO reached the minimum measured COF of 0.01. The friction values of PEG started around 0.15 and decreased to 0.08. In both cases, measurement deviations between the test runs were very low.

The WSD are in agreement with the measured friction values. The lowest diameter of $233\,\mu\text{m}$ was measured for PEG. Although its COF is higher compared to rapeseed oil, lower wear indicates an overall better separation of the tribological surfaces. Surprisingly, the wear scar diameters of rapeseed oil ($352\,\mu\text{m}$) and additivated water ($347\,\mu\text{m}$) were in close agreement. The low molecular weight derivatives C2XS and C7XS resulted in elevated wear scar diameters, irrespective of the viscosity grade. Addition of C9M and C9S had no clear effect on wear. Relatively low friction and wear of water is due to the addition of the antiwear additive. Comparative measurements without additivation showed substantially higher wear and friction.

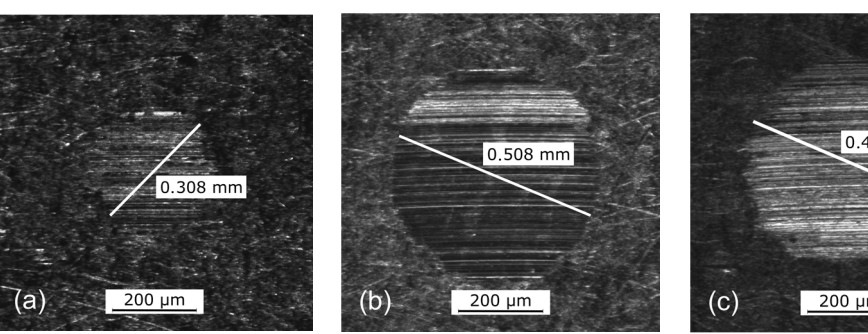

**Figure 10.** Optical images of the wear scar diameter (WSD) at ZSVG 46 for (**a**) C9L (**b**) C2XS and (**c**) C7XS (ball-on-three-plates, $F = 10\,\text{N}$, $T = 40\,^\circ\text{C}$).

Minimum friction values of 0.08 for the CMC solutions indicate that boundary or mixed lubrication is observed for all biopolymer samples. A clear decline in friction due to elastohydrodynamic film formation was not detected in any case. On the contrary, zigzag deviations or leaps at sliding velocities above $1000\,\text{mm}\,\text{s}^{-1}$. point to unstable lubricating films and contact of the steel surface asperities. Preliminary experiments, without the addition of antiwear additive, support this assumption. In all cases, high sliding speeds resulted in severe scuffing.

Assuming initial Hertzian pressure, the minimum lubricating film thickness $h_{min}$ was calculated at maximum sliding velocity $u_s$ of $1.4\,\text{ms}^{-1}$ and infinite dynamic viscosity $\eta_\infty$, using Equation (10). According to Section 3.3, the pressure-viscosity coefficients $\alpha$ were $1.65\,\text{GPa}^{-1}$ at ZSVG 46 and $1.88\,\text{GPa}^{-1}$ at ZSVG 220. Due to the lower solution concentrations of C9S, C9M and C9L compared to C7XS, the pressure-viscosity coefficients $\alpha$ and thus minimum film thicknesses are likely to be lower than estimated. Wear of the tribological surfaces during the measurements would decrease the initial Hertzian pressure and thus increase the lubricating film thickness. The corresponding lambda ratios $\lambda$ were calculated at initial surface roughness $R_a$ using Equation (9). The minimum lubricating film thickness $h_{min}$ and lambda ratio $\lambda$ of all tested solutions are listed in Table 4.

All the relevant parameters and procedure for calculating $h_{min}$ and $\lambda$ are provided in Supplementary Materials.

The maximum lambda ratio $\lambda$ of 2.03 for C2XS confirms the assumption of boundary and mixed lubrication ($\lambda < 3$), even at high sliding speeds [44]. Lambda ratios $\lambda$ above 3, assigned to elastohydrodynamic lubrication have not been achieved. Declining friction values for C9L and C9M are attributed to an increasing alignment of the polymers in sliding direction and the associated reduction in flow resistance. The lower COF of CL9 at high sliding speeds and the decrease in WSD by up to 15% compared to AW indicate absorbed boundary layers. Measurements by Guan et al. [36] showed a similar improvement in WSD of up to 18% at a concentration of 0.7 wt% CMC. Concentrations above 0.7 wt% also led to an increase in WSD. Gelinski et al. [23] were able to reduce the wear ellipsis from 27.5 to 15.5 mm$^{-2}$, by the addition of up to 0.8% carboxymethyl chitosan. Experimental studies by Naik et al. [13] showed decreasing WSD with increasing hydroxyethyl cellulose concentration and decreasing molecular weight $M_W$. Increasing friction values in boundary lubrication have previously been observed by Campen et al. [75] for closed packed films of organic friction modifiers. According to the authors, increasing friction with speed is the consequence of sliding between ordered densely packed monolayers. Furthermore, investigations of Liu et al. [76] suggest that the adverse effects of C2XS are due to high interactions and entanglement of the CMC polymers. According to their results, polymers with radii larger than half the height of the lubricating gap are influenced by the confinement, and viscous layers of adsorbed polymers hinder the access of subsequent molecules. With an estimated gap height of 10 nm for aqueous solutions, the maximum unaffected polymer radius is around 5 nm [76]. The polymer radius of a 90 kDa CMC is around 16 nm in 0.1 M NaCl solution and ∼40 nm in pure water, thus considerably higher than 5 nm [77]. Increasing COF for C2XS at ZSVG 46 and scuffing for C2XS and C7XS at ZSVG 220 indicate that the chain lengths of both molecules are too large for them to enter the lubricating gap.

Further increase of the zero-shear viscosity by increasing the concentration did not lead to an improvement of elastohydrodynamic film formation. In conclusion, achieving elastohydrodynamic lubrication under sliding conditions is not possible with the commercially available CMC derivatives studied and an increase in minimum lubricating film thickness, would require smaller molecules, with polymer radii below 5 nm.

*3.7. Lubricating Performance under Rolling Conditions*

Figure 11a,b show the coefficient of friction as a function of speed at 30 N normal load and a SRR of 30% at 40 and 60 °C, respectively. In all cases, friction decreased with sliding speed.

Independently of the measurement temperature, the COF of C7XS was about 35% lower, compared to the other tested derivatives, reaching a minimum value of 0.01, at a maximum speed of 3500 mm s$^{-1}$. As measurements were conducted at decreasing velocity, slight fluctuations at high speeds might be explained by running-in behavior and polymer orientation. In comparison to pure sliding, rolling seems to hinder polymer entanglement, improve the polymer intake into the lubricating gap and thus enhance film formation. Additionally, the higher entertainment speed $u_E$ of maximum 3500 mm s$^{-1}$ results in a minimum lubricating film thickness of around 230 nm, corresponding to a lambda ratio $\lambda$ above 5. Therefore, the limit value of 6 kDa, reported by Liu et al. [76] is primarily applicable to sliding conditions. In rolling motion, polymers with average molecular weights around 25 kDa clearly decrease friction and improve the lubricating performance. Slightly lower friction values at 60 °C compared to 40 °C are attributable to the lower solution viscosity at higher temperature. In comparison, COF measured by Sagraloff et al. [9] for aqueous biopolymer solutions at 40 and 60 °C and Stribeck curves published by Vengudusamy et al. [78] for synthetic and mineral oils gear oils at 100 °C were on a comparable level.

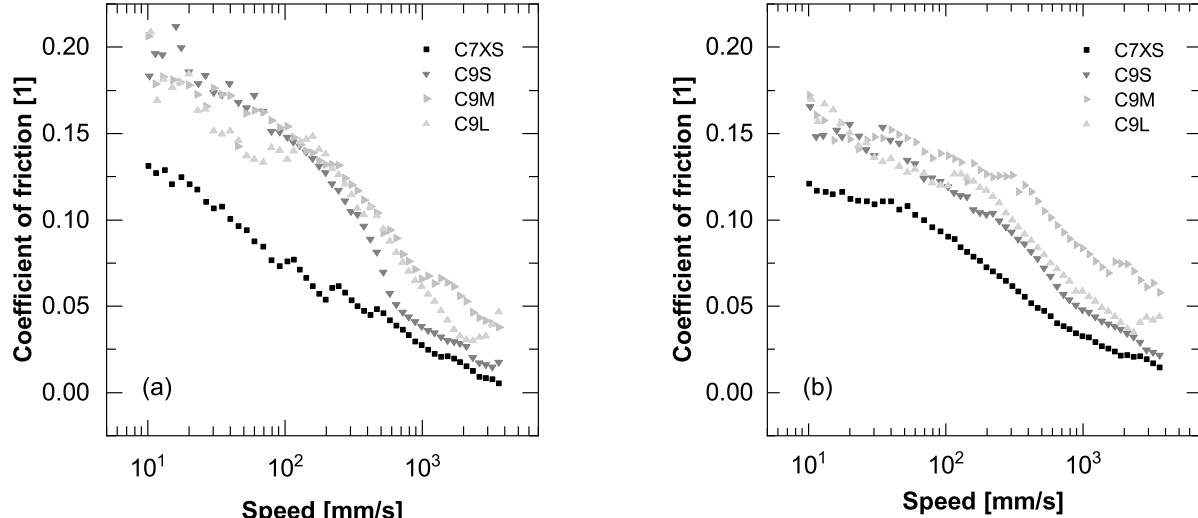

**Figure 11.** Coefficient of friction as a function of speed for C7XS, C9S, C9M and C9L at (**a**) 40 °C (**b**) 60 °C (ball-on-disc, SRR = 30%).

*3.8. Permanent Viscosity Loss*

Permanent viscosity loss (PVL) of the fully additivated CMC solutions, already applied in Section 3.7, were measured in a tapered roller bearing shearing cell. The required loading duration was determined in preliminary tests with a high molecular derivative and set to 24 h (data not shown). Figure 12a,b shows the flow curves for the fresh and sheared polymer solutions, respectively. Table 5 lists zero viscosities of the fresh ($\eta_{0,0}$) and sheared ($\eta_{0,24}$) fluids, calculated PVL and concentrations of all investigated polymer solutions.

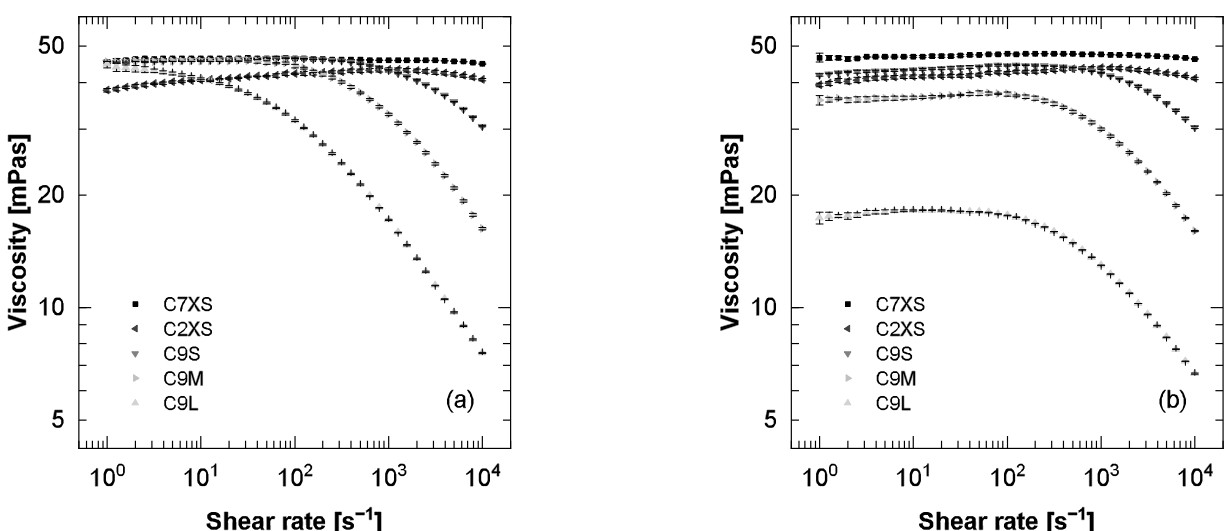

**Figure 12.** Double logarithmic plot of the viscosity as a function of shear rate for the (**a**) fresh and (**b**) sheared CMC solutions.

While viscosities for C7XS and C2XS show no detectable difference, C9M und C9L clearly deviate downwards. Overall, permanent viscosity losses increased with average molecular weight $M_W$ and decreased with concentration, which corresponds to the results of Antti et al. [79] and Almeida and Dias [80]. Measurements by Schweiger [81] showed similar viscosity losses of up to 70% for high molecular cellulose derivatives. In theory, shear forces are highest in the middle region of the polymers and molecules break into two roughly equal parts. Below a limiting molecular weight, shear forces are insufficient to

further fragment the molecules. The smallest fragments are hence a little bigger than half of the limiting molecular weight [7,46] Constant zero-shear viscosities and flow curves of C7XS and C2XS are indicative of shear stability. Molecular weights of both derivatives are below the limiting molecular weight. All other CMC solutions were affected by shear and showed PVL between 6.21 and 58.75%. Consequently, the limiting molecular weight was between 35 (C2XS) and 88 kDa (C9S) under the given shearing conditions. Higher loads and shearing energies, as stated in the current standard, would probably further reduce the limiting value. To conclude, usage of low molecular weight derivatives below the limiting molecular weight are preferable in high shear gear and bearing applications applications. Naik et al. loss of 19.4% in Kurt Orbahn rig test, HEC average molecular weight of 45 KDa. stabalized at a molar mass of $\sim$80 kDa temperatures of 25 to 40 °C [23].

**Table 5.** Zero-shear viscosity of the fresh $\eta_{0,0}$ and sheared $\eta_{0,24}$ solutions, permanent viscosity loss PVL and polymer concentration $c_{CMC}$.

| Derivative | $M_W$ [kg/mol] | $\eta_{0,0}$ [mPas] | $\eta_{0,24}$ [mPas] | PVL [%] | $c_{CMC}$ [wt%] |
|---|---|---|---|---|---|
| C7XS | 24 * | 46.44 | 47.68 | - | 10.68 |
| C2XS | 35 * | 43.21 | 43.75 | - | 7.94 |
| C9S | 88 * | 45.70 | 42.86 | 6.21 | 3.30 |
| C9M | 240 * | 45.64 | 37.10 | 18.71 | 1.17 |
| C9L | 520 * | 45.81 | 18.89 | 58.76 | 0.34 |

* estimated according to Miehle et al. [48].

## 4. Conclusions

In this study, the rheological and tribological properties of aqueous CMC solutions and the effect of concentration and molecular weight were studied to assess the potential of sodium carboxymethyl celluloses as thickening agent in sustainable, aqueous lubricants. The results were compared to rapeseed oil, polyethylene glycol 400 and additivated water, serving as biodegradable reference lubricants. The solution viscosities were adjustable to all viscosity grades relevant in gear and bearing applications. Viscosity indices are similar or better compared to the reference fluids. The rheological investigations showed a clear increase in temporary and permanent viscosity loss with molecular weight. Low DS promoted thixotropic behavior and lowered temperature stability. Tribological measurements in a ball-on-three-plates system displayed no improvement in elastohydrodynamic film formation under sliding conditions by the addition of carboxymethyl cellulose. The measurement results indicate that the average molecular weight of all derivatives used was too high for the molecules to enter the lubricating gap. Under rolling conditions, low molecular weight derivatives reduced friction by up to 35%. In order to achieve Newtonian flow behavior, increase the pressure-viscosity coefficient and improve elastohydrodynamic film formation future investigations should focus on lower molecular weight derivatives and higher concentrations.

**Supplementary Materials:** The following supporting information can be downloaded at: https://www.mdpi.com/article/10.3390/lubricants11030112/s1.

**Author Contributions:** Conceptualization, J.U.M. and S.K.; methodology, J.U.M.; formal analysis, J.U.M. and S.K.; investigation, J.U.M.; resources, T.A. (Tobias Amann), C.F. and T.A. (Tobias Asam); writing—original draft preparation, J.U.M. and T.A. (Tobias Amann); writing—review and editing, S.K., T.A. (Tobias Asam), C.F. and P.E.; visualization, J.U.M., S.K. and T.A. (Tobias Amann); supervision, P.E.; project administration, J.U.M., S.K. and P.E. All authors have read and agreed to the published version of the manuscript.

**Funding:** This research was partly funded by the Bavarian Research Foundation grant number Az-1314-17.

**Data Availability Statement:** The data presented in this study are available on request from the corresponding author. The data are not publicly available due to privacy reason.

**Acknowledgments:** The authors thank Carl Bechem GmbH, Torqeedo GmbH, Leistritz Pumpen GmbH, Renk AG, Wittenstein alpha GmbH and Forschungsstelle für Zahnräder und Getriebesysteme (FZG) for their participation in the research project.

**Conflicts of Interest:** The authors declare no conflict of interest.

## Abbreviations

The following abbreviations are used in this manuscript:

AW       Additivated Water
COF      Coefficient of Friction
CMC     Sodium Carboxymethyl Cellulose
DS        Degreee of Substitution
MTM     Mini Traction Machine
PEG      Polyethylene Glycol 400
PVL      Permanent Viscosity Loss
RSO      Rapeseed Oil
SRR      Slide-Roll Ratio
TVL      Temporary Viscosity Loss
VM       Viscosity Modifier
WSD     Wear Scar Diameter
ZSVG   Zero-Shear Viscosity Grade

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
