# Peer review of "Thickening Properties of Carboxymethyl Cellulose in Aqueous Lubrication"

_lubricants, doi:10.3390/lubricants11030112_

Round 1

Reviewer 1 Report

The article ‘Thickening Properties of Carboxymethyl Cellulose in Aqueous Lubrication’ submitted by Michaelis and co-workers investigated the rheological and tribological characteristics of aqueous carboxymethyl cellulose solutions as green aqueous lubricants. The effect of concentrations, molecular weights and viscosity grades on the performance of rheology and tribology was further discussed. My suggestion for this article is major revision. Some comments are as follows:

1.      Please briefly describe the carboxymethyl cellulose for what field of lubrication?

2.      Table 1 shows the viscosity range of all cellulose derivatives at 25℃ with different concentration. Are the CMC solutions mentioned in Table 1 commercially available or lab-prepared? Why are solution concentrations inconsistent?

3.      Figure 4 exemplarily shows the resulting flow curves for C9L on the left and C2XS on the right. For C2XS, there also seems to be no obvious shear thinning phenomenon even at more concentrated solutions (colored in light gray). It is suggested to provide the flow curves for other CMC samples in the supplementary document for comparison.

4.      In line 246, “At constant molecular mass, zero viscosity at a DS of 0.7 is lower compared to a DS of 0.95 … …” But there is no value for DS=0.95 in Table 2. Please check.

5.      In line 265 and line 271, “BS9” should be “B9S”.

6.      After section 3.3, the data about B9S is not listed in the article. The follow-up discussion did not include all CMC samples. What is the reason for this?

7.      It is necessary to provide relevant parameters and procedure for calculating λ value in the supplementary document.

8.      The authors mentioned in the abstract that the lubricating mechanism was studied. However, the discussion on lubricating mechanism in the manuscript was not fully reflected. It is suggested to supplement the corresponding content and further discuss the relationship between the structure and the tribological performance of carboxymethyl cellulose.

9.      The authors mentioned that improvement of the lubricating properties and film-forming capability would require smaller molecules. Besides the effect of polymer size, are there any other factors affecting the lubrication properties? For example, the distribution of polar functional groups. It is generally believed that polar functional groups contribute to the adsorption of lubricating additives on the sliding surfaces to form protective films.

Reviewer 2 Report

 The manuscript is interesting and clearly written with a logical sequence.  In my opinion, the manuscript can be acceptable in present form.  

Author Response

Dear Reviewer,

Thank you for your time and comments, which help us a lot to improve the manuscript. We have included all changes and comments below.

Reviewer 3 Report

1.      Abstract should be given as more interesting. Express at least one of the main aspects and features of the paper.

2.      Improve the conclusion part of the Abstract.

3.      Wherever applicable, the scientific explanation needs to be added and the research novelties need to be clearly emphasized.

4.      At the end of Introduction section, it would be better to add the paper's organization in different sections.

5.      Given that the manuscript is based on simulation work and measurements, it is vital for the authors to report on the method(s) to improve measurement reliability. The methods/measures that have been taken to minimize simulation errors and improve reliability should be included in the experimental setup.

6.      Further, results and analysis of experiments should be compared with previous researchers by citing references.

7.      Provide citation of literature for Eq. (2).

8.      Conclusion must be presented in highlight the contribution, and applicability of the work.

9.      Please check the manuscript for wrong choice of words, grammatical errors and incoherent sentence structure.

Reviewer 4 Report

The article has a scientific character. The article deals with Properties of Carboxymethyl Cellulose in Aqueous Lubrication. The work is part of the search for biodegradable lubricants for loaded friction nodes of machines such as gears. The Authors applied correct research methods and used the appropriate measuring equipment. The content of the work is logically written. The manuscript contains 11 figures and 5 tables. Figures and tables are properly prepared. Authors cited 79 literature sources. The authors presented an interesting work, I found no significant objections to it.

However, I have recommendations to improve the usability of the work for the reader:

- the work should also present the results in the COF system - Hersey-Stribeck number; it would be beneficial to show the areas for fluid, mixed or boundary friction,

- the tested compositions should be assigned to viscosity classes VG,

- in tests, provide results also for mineral oils for comparable VG classes

- The authors should also show the views of the surfaces after tribological tests,

- The authors provided formulas for the lambda parameter, but it was not used to interpret the results.

Round 2

Reviewer 1 Report

Please update the references if possible. I have the following suggestions:

In order to highlight the pioneering of this investigation, it is suggested to cite the literature of recent years about the progress of water-based lubrication or polyethylene glycol lubrication. For example, reference 10, 11, 12...

The format of the references in the article should be consistent, such as capitalizing the title of the reference or capitalizing the first letter of each word. For example, reference 1, 2, 3...

The format of the journals in the references should be uniform, using full names or abbreviations. For example, reference 14, 20, 23...

Reviewer 3 Report

Authors have made significant changes in the revised manuscript. Hence, consider the manuscript for publication in its present form.

Author Response

Dear Reviewer,

Thank you very much for your time and valuable comments, which helped us a lot to improve the manuscript.